WRKY transcription factor family in lettuce plant (Lactuca sativa): Genome-wide characterization, chromosome location, phylogeny structures, and expression patterns

Du Ping
Wu Qinglian
Liu Yihua
Cao Xue
Yi Wenyi
Jiao Tikun
Hu Mengqi
Huang Ying hyhappy1314@163.com
Linyi University , Linyi , China
Siddique Muhammad Hussnain
Electronic publication date: 2022 Oct 18
Publication date: 2022
Volume: 10
Electronic Location ID: e14136
Received 2022 Feb 22; Accepted 2022 Sep 6
Copyright: ©2022 Du et al.
Copyright year: 2022
Copyright holder: Du et al.
License: This is an open access article distributed under the terms of the Creative Commons Attribution License, which permits unrestricted use, distribution, reproduction and adaptation in any medium and for any purpose provided that it is properly attributed. For attribution, the original author(s), title, publication source (PeerJ) and either DOI or URL of the article must be cited.
License URL: https://creativecommons.org/licenses/by/4.0/

Keywords: Asparagus lettuce, WRKY TF, Abiotic stress, Expression patterns

Funding: National Natural Science Foundation of China 32102405 31801868 Natural Science Foundation of Shandong Province ZR2020QC156 ZR2020QC063 This work was supported by: the National Natural Science Foundation of China: 32102405, 31801868, and the Natural Science Foundation of Shandong Province: ZR2020QC156, ZR2020QC063. The funders had no role in study design, data collection and analysis, decision to publish, or preparation of the manuscript.

==============================
WRKY transcription factors (TF) have been identified in many plant species and play critical roles in multiple stages of growth and development and under various stress conditions. As one of the most popular vegetable crops, asparagus lettuce has important medicinal and nutritional value. However, study of WRKY TFs family in asparagus lettuce is limited. With the lettuce (Lactuca sativa L.) genome publication, we identified 76 WRKY TFs and analyzed structural characteristics, phylogenetic relationships, chromosomal distribution, interaction network, and expression profiles. The 76 LsWRKY TFs were phylogenetically classified as Groups I, II (IIa-IIe), and III. Cis element analysis revealed complex regulatory relationships of LsWRKY genes in response to different biological progresses. Interaction network analysis indicated that LsWRKY TFs could interact with other proteins, such as SIB (sigma factor binding protein), WRKY TFs, and MPK. The WRKYIII subfamily genes showed different expression patterns during the progress of asparagus lettuce stem enlargement. According to qRT-PCR analysis, abiotic stresses (drought, salt, low temperature, and high temperature) and phytohormone treatment could induce specific LsWRKYIII gene expression. These results will provide systematic and comprehensive information on LsWRKY TFs and lay the foundation for further clarification of the regulatory mechanism of LsWRKY, especially LsWRKYIII TFs, involved in stress response and the progress of plant growth and development.

Introduction

Long-term domestication and directional selection of lettuce have resulted in the development of various cultivars, oil lettuce, stem lettuce (also known as asparagus lettuce), and various varieties of leaf lettuce to meet various needs. Asparagus lettuce (Lactuca sativa L. 2n = 2x = 18), an annual or biennial variety of lettuce that can form fleshy tender stems, is a member of the Lactuca genus of Compositae family. In 2016, the world’s lettuce production (including chicory) and cultivation area were 26.78 million tons and 1.223 million hectares, respectively (http://www.fao.org/faostat/en/). China has the highest yield and cultivation area globally, accounting for 56% and 51% of the total, respectively. Asparagus lettuce is widely cultivated and consumed throughout the year in China’s North and South. It contains various vitamins, proteins, fats, and phytochemicals (flavonoids and terpenoids). As an economically important vegetable, asparagus lettuce plays a vital role in balancing people’s diets (Cui et al., 2014). The freshness and tenderness of stems influence the quality of asparagus lettuce. Due to advancements in molecular biology techniques, more and more technologies, such as whole-genome analysis, cell activity analysis, and linkage map analysis, can be used to investigate the molecular mechanism of stem enlargement (Li et al., 2020a). However, the regulatory mechanism of lettuce stem expansion remains unclear.

Plant transcription factors (TFs) have been found to play essential regulatory roles in stem enlargement. MADS-box, ABF/AREB, and homeobox TFs were discovered to be involved in forming roots and tubers (Pernisova, Kuderova & Hejatko, 2011). MADS-box TFs, including IbMADS1, IbMADS3, IbMADS4, and IbMADS79, were mainly expressed in the root tubers of sweet potatoes (Kim, Mizuno & Fujimura, 2002; Kim et al., 2005; Cheng et al., 2013). ABF4 (ABF-binding factor) positively regulates potato tuber induction. Overexpression of Arabidopsis ABF4 in potato improved its production as well as salt and drought tolerance (Garcia et al., 2018). The silencing of StNAC103, which was discovered in potato tuber periderm, increased the total load of suberin and wax in the periderm (Verdaguer et al., 2016). However, the roles of WRKY TFs in plant root and stem development remain unclear.

Containing the conserved WRKY domain in the N terminus and zinc finger motif in the C terminus (C2H2 or C2HC), WRKY proteins can recognize and bind to the W-box element (TTGAC/T) in the promoter region of target genes. WRKY TFs are classified into three types: Group I (two WRKY domains with C2H2 motif), II (one WRKY domain with C2H2 motif), and III (one WRKY domain with C2HC motif) (Eulgem et al., 2000). Group II is further subdivided into five subgroups, i.e., IIa-IIe. The WRKY TFs regulate multiple physiological processes (Dong, Chen & Chen, 2003; Rengasamy et al., 2008; Li et al., 2020b; Liu et al., 2021a; Wei et al., 2021). Drought stress elicited 88 WRKYs in Phaseolus vulgaris and 58 WRKYs in maize (Wu et al., 2016; Zhang et al., 2017). Brachypodium distachyon BdWRKY38 has been identified as a participant in response to Rhizoctonia solani by mediating SA signaling (Kouzai et al., 2020). Tomato WRKYIII subfamily gene SlWRKY81 inhibited plant drought tolerance by suppressing SlRBOH1-derived H2O2 accumulation (Ahammed, 2019).

WRKY TFs also involve in plant development such as seed germination, reproductive processes, senescence, and plant organ development (Chen et al., 2017). AtWRKY6 played a vital role in leaf senescence by regulating the enzyme SIRK (Robatzek & Somssich, 2002). Zhang et al. (2011) found that rice OsWRKY78 could regulate stem elongation; the expression pattern of OsWRKY78 in the elongated stem was most abundant, and inhibition of OsWRKY78 expression resulted in the shortening of somatic cell length. Cotton GhWRKY15 improved resistance to the virus, fungal infection, and stem elongation (Yu et al., 2012). Li et al. (2016) found that WRKY TFs were involved in carrot root development.

As a result of systematic studies, many WRKY TF family members have been identified in different plant species, such as 72 in Arabidopsis, 81 in tomato, 95 in carrot, 55 in cucumber, 59 in grape, 45 in Eucommia ulmoides, and 64 in Isatis indigotica (Ishiguro & Nakamura, 1994; Yang et al., 2020; Liu et al., 2021b; Qu et al., 2021). However, systematic and comprehensive information of WRKY TFs in asparagus lettuce were unclear.

In this study, 76 WRKY TFs were identified in asparagus lettuce through genome-wide analysis. Exon-intron structure, phylogenetic relationships, motif compositions, collinearity analysis, and chromosome distribution of WRKY genes were identified. WRKYIII subfamily TFs have been identified to involve in different biological processes. So, we investigated the expression profiles of WRKYIII genes at different stages of stem expansion, various abiotic stresses, and plant hormones. Our results will provide the basis of WRKY TFs in asparagus lettuce and will highlight the role of WRKY TFs, especially WRKYIII TFs, in stem expansion and stress response.

Material and Methods

Sequence retrieval and identification of WRKY TFs in lettuce

The lettuce WRKY TFs were obtained from the lettuce’s genome (https://lgr.genomecenter.ucdavis.edu/). Amino acid sequences of Arabidopsis WRKY TFs were used as query sequences to search for the homologous LsWRKY TFs sequences. Subsequently, the conserved WRKY domain was identified by SMART (http://smart.embl-heidelberg.de/), Pfam database (http://pfam.janelia.org/), and NCBI CDD search (https://www.ncbi.nlm.nih.gov/Structure/cdd/wrpsb.cgi). The molecular weight (Mw) and theoretical isoelectric point (pI) of LsWRKY TFs were identified by ExPASY server (http://www.expasy.ch/tools/pi_tool.html).

Gene structure, conserved motif, and cis-elements analysis of promoter

GSDS (http://gsds.gao-lab.org/) was used to analyze the structure of LsWRKY TFs. MEME online program (https://meme-suite.org/meme/tools/meme) was used to identify the conserved motif of LsWRKY TFs (Bailey et al., 2009). The result was visualized by TBtools software (Chen et al., 2020). To investigate cis-elements, the promoter region of 2000 bp genomic DNA upstream sequence was submitted to the PlantCARE database (Lescot et al., 2002).

Multiple sequence alignment and phylogenetic tree of LsWRKY TFs

Multiple sequence alignment of LsWRKY TFs was performed using the DNAMAN software. The WRKY TFs of Arabidopsis and tomato were downloaded from TAIR (https://www.arabidopsis.org/) and SGN (https://solgenomics.net/), respectively. The phylogenetic tree was constructed by using MEGA 7.0 software with the neighbor-joining method with 1,000 bootstrap replicates (Kumar et al., 2008).

Chromosomal distribution and syntenic relationship of LsWRKY TFs

MapChart software was used to draw the chromosomal distribution of WRKY TFs from lettuce’s genome, while STRING software was used to conduct the interaction network (Franceschini et al., 2013). McscanX was used to identify the orthologous and paralogous genes of WRKY TFs in L. sativa and willow-leaf lettuce L. saligna to elucidate LsWRKY TFs’ origin. The symbiotic relationships were displayed using Circos software (Krzywinski et al., 2009).

Plant materials, stress and phytohormone treatments

Seeds of the cultivated asparagus lettuce ‘Yonganhong’ were sown in a controlled environment chamber for 12 h photoperiod at 22 and 18 °C (day vs. night) with a light intensity of 20,000 µmol/m2/s (lux). Asparagus lettuce seedlings were used in subsequent experiments once they reached the four-leaf stage. Seedlings were treated with 200 mmol/L NaCl (salt), 20% PEG6000 (drought), 4 °C (low temperature), and 37 °C (high temperature) for abiotic stress treatment, respectively. Salicylic acid (SA, 0.5 mmol/L), abscisic acid (ABA, 75 µmol/L), and gibberellin (GA, 50 µmol/L) were sprayed on them and placed for different time. The expression patterns of LsWRKY TFs were also analyzed at different stages of asparagus lettuce stem development (S1: the transverse diameter length is one cm, S2: the transverse diameter length is two cm, S3: the transverse diameter length is three cm, and S4: the transverse diameter length is four cm) (Fig. 1). All samples were frozen in liquid nitrogen and stored in a −80 °C refrigerator. Total RNA of different treatment including 4 stem swelling stages (S1, S2, S3, and S4), abiotic stress (drought, salt, low temperature, and high temperature), and hormone treatment (ABA, SA, and GA) was isolated using a plant total RNA isolation kit (Vazyme, Nanjing, China) and first-strand cDNA was synthesized using a 1st Strand cDNA Synthesis Kit (Vazyme, Nanjing, China).

Quantitative transcript analysis and qRT-PCR validation

Four stem enlargement stages (S1, S2, S3, and S4) of ‘Yonganhong’ were chosen to conduct an RNA sequencing to identify the differentially expressed genes. Each stage contained three biological replicates for the RNA sequencing. PCR amplified the purified double-stranded cDNA were further purified to obtain libraries for Illumina HiSeq 4000 sequencing (Biomarker Technologies Co., Ltd., Beijing, China). The transcript abundance of LsWRKY genes in different development stages was counted by HTSeq and FPKM (fragments per kilobase exon per million fragments mapped) to estimate the expression level. The statistical power of the identified genes in this experimental design was calculated by in Biomarker Technologies Co., Ltd., (Beijing, China) referred to the RNASeqPower software (https://doi.org/doi:10.18129/B9.bioc.RNASeqPower) (Hart et al., 2013). The transcriptome sequencing data has been submitted to a public database (NCBI: PRJNA844256).

Figure 1 The cross-sections of the asparagus lettuce stem development stages.

S1, the transverse diameter is one cm; S2, the transverse diameter is two cm; S3, the transverse diameter length is three cm; S4, the transverse diameter length is four cm. Scale bars: 0.5 cm (S1); one cm (S2, S3, S4).

qRT-PCR was conducted to verify the expression patterns of LsWRKY genes. For qRT-PCR, SYBR Green I (TaKaRa, Dalian, China) and the Roche LightCycler 96 were used. LsTIP41 (Lsat_1_v5_gn_5_116421) was used to normalize and calculate the expression levels of each LsWRKYIII gene (Borowski et al., 2014). The relative expression levels of LsWRKYIII genes were calculated using the 2−ΔΔCT methods based on the mean value of three technical repeats. The primer pairs were designed by Primer Premier 6.0 and are listed in Table S1.

Statistical analysis

For experimental data analyses, ANOVA (an analysis of variance) was performed by using SPSS software (version 17.0; SPSS, Inc., Chicago, IL, USA) to investigate whether there was a significant difference among the tested samples.

Results

Identification of LsWRKY TFs in lettuce

In the earlier research, Guo et al. (2019) has identified 74 WRKY TFs in lettuce. In this study, a total of 76 LsWRKY TFs (denoted as LsWRKY01 to LsWRKY76) were identified, which increased two TFs i.e., LsWRKY02 (Lsat_1_v5_gn_0_3061.1) and LsWRKY38 (Lsat_1_v5_gn_6_220.3). The coding sequence (CDS) lengths of LsWRKY TFs ranged from 546 bp (LsWRKY02) to 2232 bp (LsWRKY42), with corresponding amino acid (aa) numbers ranging from 181 aa to 743 aa. The MWs and pI values of the identified LsWRKY TFs ranged from 20.7 kDa (LsWRKY02) to 81.7 kDa (LsWRKY42) and from 5.19 (LsWRKY44) to 9.98 (LsWRKY35), respectively. The polypeptide was composed of 59.90% aliphatic amino acids and 7.50% aromatic amino acids. The GRAVY values ranged from −1.274 to −0.46, indicating that LsWRKY proteins are hydrophilic (Table S2).

Multiple sequence alignment and phylogenetic analysis of LsWRKY TFs

As shown in Fig. S1, sequence alignment of LsWRKY TFs was conducted. Two WRKY domains with the conserved WRKYGQK were present in Group I, which contained C2H2-type zinc-finger domains. All 46 LsWRKY TFs in Group II, such as LsWRKY03, LsWRKY05, LsWRKY10, and LsWRKY13, had one WRKY domain and a C2H2-type zinc-finger. Members in Group III had one complete WRKY domain and a C2HC zinc finger. However, the WRKYGQK sequence was found changed in some LsWRKY TFs, i.e., WRKYGKK in LsWRKY50 and WKKYGEK in LsWRKY61.

To investigate the phylogenetic relationship of LsWRKY TFs, the phylogenetic tree conducted with amino acid sequences of WRKY TFs from different plant species was also constructed using MEGA7.0 software (Fig. 2). For lettuce WRKY TFs, Group II had the most members (46), but the distribution was uneven among the five subgroups IIa (3), IIb (9), IIc (17), IId (9), and IIe (8). Group I contained 17 LsWRKY TFs, while Group III formed the smallest group with 13 LsWRKY TFs (Fig. S2 and Table S2). A total of 949 WRKY TFs from 10 different plant species were chosen to analyze the classification of the WRKY TFs family (Fig. S3). Glycine max had the most WRKY TFs (176), followed by Zea mays (131), Oryza sativa (100), and Daucus carota (95); Vitis vinifera had the fewest WRKY TFs (59). Among the three groups, WRKY TFs were mainly classified into Group II. For instance, 45 of 72 A. thaliana WRKY TFs belonged to Group II, while Groups I and III contained 13 and 14 members. The distribution of WRKY TFs in Solanum lycopersicum was 15 (Group I), 52 (Group II), and 11 (Group III).

Figure 2 Phylogenetic analysis of WRKY TFs among lettuce, Arabidopsis, and tomato plants.

MEGA 7.0 software was used to conduct the phylogenetic analysis using a neighbor-joining method with 1,000 bootstrap replicates. Different colors represent different subfamily WRKY TFs.

Gene structure, conserved motif, and cis-elements analysis of LsWRKY TFs

The GSDS program was used to analyze the introns and exons of LsWRKY TFs. The numbers of introns in 76 LsWRKY TFs ranged from one to six. The majority of members (33 of 76 LsWRKY TFs) had two introns and three exons, followed by three introns (17) and four introns (10). The LsWRKY52 had the highest number of introns (six) and exons (seven), while LsWRKY73, LsWRKY29, LsWRKY12, LsWRKY30, and LsWRKY58 both had only one intron (Fig. 3B). The losses and gains of LsWRKY TFs may be related to the functional diversity during the evolution of LsWRKY TFs.

Figure 3 Phylogenetic relationship, exon-intron structure, and conserved motifs analysis of WRKY TFs in lettuce.

(A) The phylogenetic tree created by MEGA 7.0 and conserved motifs predicted in WRKY protein. The MEME program identified the ten motifs, with each number of the colored box representing a different motif. (B) Exon-intron structures from online software GSDS. Yellow boxes and lines represent exons and introns, respectively; blue boxes represent the UTR.

Despite the gene structure of LsWRKY TFs differed, some conserved motifs were found in all LsWRKY TFs. MEME program identified ten conserved motifs to illustrate the similarity and diversity of motif composition. The conserved motifs in 76 LsWRKY TFs ranged from two to seven. Motif 1 and motif 2 existed in all 76 LsWRKY TFs. There were only two motifs in nine LsWRKY TFs (LsWRKY51, 60, 69, 26, 07, 44, 66, 67, and 73). Motif 9 and motif 10 mainly existed in Group III and II, respectively. Motif 3 and motif 5 were unique in Group I, such as LsWRKY65, 32, 24, 53, 74, 06, 59, 42, and 48 (Fig. 3A). The results indicated that LsWRKY TFs from the same group have similar conserved motifs. But differences also existed in these LsWRKY TFs, indicating the functional diversity of LsWRKY TFs (Rose, 2004).

To conduct the cis-elements analysis, 2.0 kb DNA sequences upstream from 76 LsWRKY TF codons were chosen. As shown in Fig. 4. 10 types of cis-elements containing hormone-related (GA-responsive element TCTGTTG, SA responsiveness element CCATCTTTTT, ABA responsiveness element ACGTG, auxin-responsive element), stress-related (defense and stress responsiveness element, low-temperature responsiveness element) and plant growth and development-related cis-elements were found in 47 LsWRKY TFs. MYB binding site (CAACAG) was found in 47 LsWRKY TFs. GA-responsive, SA-responsive, and ABA-responsive elements were found in 70, 32, and 56 LsWRKY, respectively. Low-temperature responsiveness element LTR (CCGAAA) was found in 23 LsWRKY TFs.

Figure 4 Cis-element analysis of LsWRKY genes in lettuce.

Boxes with different colors represent different cis-element identified by the PlantCARE program, with each number of the colored box representing a different motif.

Chromosomal distribution and syntenic relationship of LsWRKY TFs

LsWRKY TFs were investigated according to the lettuce genome database to evaluate the chromosomal distribution. Except for LsWRKY02, which was located on unanchored contig, 75 LsWRKY TFs were identified to locate on nine lettuce chromosomes (Fig. 5). The LsWRKY TFs were mainly found on chromosome 09 (15), followed by chromosome 07 (13), chromosome 04 (11), and chromosome 08 (10). The number of LsWRKY TFs was seven on both chromosomes 3 and 5. Four LsWRKY TFs were found on chromosome 6. Only 3 LsWRKY TFs were mapped on chromosome 01. L. saligna, which also belonged to the genus Lactuca, was chosen to construct the comparative analysis to identify the paralogs and orthologs with L. sativa. As shown in Fig. 6, 75 and 70 pairs of paralogs were identified in L. sativa and L. saligna, respectively. Moreover, 75 pairs of orthologs between L. sativa and L. saligna were identified (Table S3).

Figure 5 Chromosomal distribution of LsWRKY TFs in the lettuce chromosomes.

Red represented the distribution of WRKYIII subfamily TFs in the lettuce chromosomes.

Figure 6 Comparative analysis of synteny between Lactuca sativa and Lactuca saligna (least lettuce, willow-leaf lettuce).

Red line represented the identified gene pair. The chromosomes marked in gray represent the chromosomes of Lactuca saligna, the identified genes belonged to Lactuca saligna are represented by LOCUSxxxxx. The color labeled chromosomes represent the chromosomes of Lactuca sativa, the identified genes belonged to Lactuca saligna are represented by LsWRKYs.

Interaction network analysis of LsWRKY TFs

To analyze the regulation mechanism, STRING software was used to construct an interaction network of LsWRKY TFs based on the orthologs in A. thaliana. As shown in Fig. 7, 49 LsWRKY TFs showed interaction with other proteins. Six lettuce LsWRKY TFs (LsWRKY14/30/31/53/65/68) were identified as the homology to WRKY33; four lettuce LsWRKY TFs (LsWRKY02/14/19/54) showed high similar to WRKY75.

Figure 7 An interaction network analysis of LsWRKY TFs.

Edges represents protein-protein associations. Different colored horizontal lines represent different interactions (known interactions, Predicted Interactions, and others).

LsWRKY TFs showed complex interaction with other proteins such as WRKY TFs, MPK4, and sigma factor binding protein (SIB). For Group III, WRKY TFs, WRKY53 (LsWRKY39/51/58) and WRKY33 (LsWRKY14/30/31/53/65/68) showed interactions with 11 or 14 stress related-proteins, including MPK proteins (MPK4 and MPK3), SIB proteins, and ACS6 respectively, indicating the critical roles in the regulation of transcription and biological processes of lettuce stem. LsWRKY8/69/75 (WRKY70) and LsWRKY14/30/31/53/65/68 (WRKY33) interacted with other proteins in a similar manner. They both interacted with SIB1, WRKY18, MPK4, and LsWRKY28, indicating that their regulatory networks were similar. Co-expression relationships existed in LsWRKY TFs and other WRKY TFs included LsWRKY13 (WRKY80), LsWRKY28 (WRKY40), and LsWRKY34/27 (WRKY60), which indicated the auto-regulation or cross-regulation with each other.

Expression patterns of LsWRKYIII subfamily genes in response to abiotic stress

To investigate the role of WRKYIII genes during abiotic stress, ten genes (LsWRKY9, LsWRKY12, LsWRKY39, LsWRKY51, LsWRKY58, LsWRKY69, LsWRKY70, LsWRKY71, LsWRKY72, and LsWRKY75) were chosen to identify the expression patterns during different abiotic stresses (salt, drought, low temperature, and high temperature) by qRT-PCR (Fig. 8).

Figure 8 Relative expression of LsWRKYIII genes under different abiotic stresses.

(A) Expression levels of LsWRKYIII genes under salt (200 mmol/L NaCl), drought (20% PEG6000), low temperature (4 °C) and high temperature (37 °C) and untreated control (CK) at 12 h; (B) expression levels of LsWRKYIII genes under salt (200 mmol/L NaCl), drought (20% PEG6000), low temperature (4 °C) and high temperature (37 °C) and untreated control (CK) at 24 h. Bars with different lowercase letters were significantly different by Duncan’s multiple range tests (p = 0.05).

Salt stress

Expression levels of LsWRKY58 were increased about two times (12 h) and ten times (24 h) after NaCl treatment. Five WRKYIII genes (LsWRKY09, LsWRKY70, LsWRKY71, LsWRKY72, and LsWRKY69) showed down-regulation expression patterns after 12 h and 24 h. Expression profiles of LsWRKY12, LsWRKY39, LsWRKY51, and LsWRKY75 were increased at 12 h but decreased at 24 h.

Drought stress

The expression profiles of LsWRKY12, LsWRKY39, LsWRKY51, and LsWRKY75 were similar under drought stress; these four genes showed increased expression levels not only at 12 h but also at 24 h. Compared with untreated control, the expression levels of LsWRKY12 and LsWRKY39 increased about two times (12 h) and five times (12 h), respectively. In contrast, LsWRKY71 showed decreased expression profiles under drought treatment, indicating that LsWRKY71 may play a negative regulatory role. LsWRKY09, LsWRKY70, and LsWRKY72 showed increased expression at 12 h but decreased expression at 24 h. There were no apparent changes in mRNA levels of LsWRKY58 at 24 h, although the expression level increased about three times at 12 h.

Low temperature (4 °C)

Under different treatment times, low temperature significantly induced the expression of 2 LsWRKY genes, i.e., LsWRKY58 and LsWRKY39. The LsWRKY58 and LsWRKY39 showed 5-fold (24 h) and 9-fold (12 h) increase, respectively. In contrast, five genes, LsWRKY12, LsWRKY69, LsWRKY70, LsWRKY72, and LsWRKY75, were up-regulated expressed at 12 h and down-regulated at 24 h, respectively. LsWRKY09, LsWRKY51, and LsWRKY71 showed decreased expression profiles throughout the treatment period.

High temperature (37 °C)

Compared with untreated control, 4 WRKYIII genes, including LsWRKY09, LsWRKY51, LsWRKY72, and LsWRKY71, showed a significantly decreased expression at 12 h and 24 h, whereas the expression of LsWRKY75 increased about three times at 12 and 24 h (Fig. 8). No significant changes were detected in the expression of LsWRKY12 (24 h) and LsWRKY58 (12 h). The expression profiles of LsWRKY39, LsWRKY69, and LsWRKY70 decreased at 12 h but increased at 24 h.

Expression patterns of LsWRKYIII subfamily genes under treatment with phytohormone

As important phytohormones, ABA, GA, and SA play critical roles in plant growth and development and various stresses by participating in various signal transduction pathways. Interestingly, most LsWRKY’s, including LsWRKYIII gene promoters, contained one or more phytohormone response elements (ABA, GA, SA, and auxin) as determined by cis- elements analysis (Fig. 4). The results indicated that LsWRKY TFs might respond to different biological progress by participating in plant hormone signaling. To investigate possible response mechanisms, we examined expression profiles of LsWRKYIII genes in response to exogenous ABA, SA, and GA.

As shown in Fig. 9, exogenous ABA, SA, and GA significantly induced the expression of LsWRKY09, LsWRKY69, and LsWRKY75 except for ABA treatment at 24 h, which showed no significant change in expression. The expression levels of LsWRKY69 and LsWRKY75 peaked under SA treatment at 24 h. LsWRKY70 and LsWRKY72 increased significantly under GA and SA treatment, while the expression levels were decreased under ABA treatment. LsWRKY12 and LsWRKY39 were induced by SA treatment, but they both showed insensitive expression patterns under ABA treatment. In construct, the expression profiles of LsWRKY58 were increased only under GA treatment at 12 h.

Figure 9 Relative expression of LsWRKYIII genes under hormone treatment.

(A) Expression levels of LsWRKYIII genes under SA (salicylic acid, 0.5 mmol/L), ABA (abscisic acid, 75 µmol/L), GA (gibberellin, 50 µmol/L) and untreated control (CK) at 12 h; (B) expression levels of LsWRKYIII genes under SA (salicylic acid, 0.5 mmol/L), ABA (abscisic acid, 75 µmol/L), GA (gibberellin, 50 µmol/L) and untreated control (CK) at 24 h. Bars with different lowercase letters were significantly different by Duncan’s multiple range tests at 0.05 levels.

Tissue-specific expression patterns of LsWRKYIII subfamily genes

To investigate the potential functions of LsWRKY TFs during the development of L. sativa, the expression patterns of 10 LsWRKYIII genes in different organs (root, stem, and leaf) were identified (Fig. 10). The expression patterns of three LsWRKY genes (LsWRKY39, LsWRKY58, and LsWRKY71) were similar; these genes showed the highest expression levels in root as compared with stem and leaf. In contrast, the expression pattern of LsWRKY12, LsWRKY72, LsWRKY51, LsWRKY69, and LsWRKY70 in the leaf and stem increased compared with the root. LsWRKY09 showed the highest expression in the leaf, while LsWRKY75 showed the highest expression in the stem. The preferential expression patterns of LsWRKYIII genes in different organs indicated that each LsWRKYIII genes might play a unique role in organ development or function.

Figure 10 Relative expression of LsWRKYIII genes at root, stem, and leaf.

Bars with different lowercase letters were significantly different by Duncan’s multiple range tests at 0.05 levels.

Transcript abundance analysis of LsWRKY genes in lettuce

RNA-seq was conducted at four points (S1, S2, S3, and S4) during the progress of stem lettuce enlargement. Three independent biological replicates of stem at each point were analyzed (12 samples each in total). A total of 89.13 Gb clean data were obtained and over 30,000 genes were identified. The statistical power ranging from 0.05 to 0.91 of each identified gene under two different points (S1 vs S2, S1 vs S3, S1 vs S4, S2 vs S3, S2 vs S4, S3 vs S4) was shown in Table S4. The developmental expression profiles of LsWRKY genes were conducted to explore the WRKY TFs function involved in the progression of stem development. As shown in Fig. 11, 43 of 76 LsWRKY TFs showed different expressions. Several genes, including LsWRKY53, LsWRKY49, LsWRKY21, LsWRKY28, LsWRKY39, and LsWRKY58, showed up-regulated expression profiles as lettuce stem enlargement progression. While, the expression levels of some genes such as LsWRKY17, LsWRKY72, LsWRKY66, and LsWRKY08 were decreased, indicating that LsWRKY genes may play negative regulatory roles in lettuce stem enlargement. Some genes showed wavy expression patterns, including LsWRKY60, LsWRKY14, and LsWRKY02, while the LsWRKYIII genes showed different expressions in the progression of stem development. For WRKYIII subfamily genes, 6 (LsWRKY39, LsWRKY58, LsWRKY69, LsWRKY70, LsWRKY71, and LsWRKY72) of 13 WRKYIII genes were detected with different expression. The expression levels of LsWRKY39, LsWRKY70, LsWRKY71, and LsWRKY58 were significantly induced, while, LsWRKY72 and LsWRKY69 showed decreased expression patterns.

Figure 11 Expression profiles of LsWRKY genes by the transcriptome data analysis at different lettuce stem enlargement periods.

S1, diameter length is one cm; S2, diameter length is two cm; S3, diameter length is three cm; S4, diameter length is four cm. a b c represents three biological replicates of each developmental stage. FPKM values of LsWRKY genes were transformed by log10(FPKM + 1), and the heatmap was constructed with TBtools software. The red box represents lettuce WRKYIII genes.

Validation of expression profile of LsWRKYIII subfamily genes at different stages of stem enlargement

To explore the function of LsWRKY genes involved in lettuce stem development, qRT-PCR was used to examine the expression profiles of 10 LsWRKYIII genes (LsWRKY09, LsWRKY12, LsWRKY39, LsWRKY51, LsWRKY58, LsWRKY69, LsWRKY70, LsWRKY71, LsWRKY72, and LsWRKY75) (Fig. 12). LsWRKY69 and LsWRKY72 showed decreased expression profiles during lettuce stem development. The results were consistent with the results of RNA-Seq. Four WRKY genes (LsWRKY58, LsWRKY70, LsWRKY39, and LsWRKY71) showed the highest expression levels at the S3 stage. Interestingly, LsWRKY09, LsWRKY12, LsWRKY75, and LsWRKY51, which RNA-Seq did not detect, also showed the response to asparagus lettuce stem enlargement by RT-qPCR. As shown in Fig. 12, the expression profiles of LsWRKY12 and LsWRKY75 both showed the highest expression levels at the S3 stage; the expression at S2 and S4 stages were decreased compared with the S1 stage. LsWRKY09 and LsWRKY51 showed opposite expression patterns. The expression levels of LsWRKY09 increased continuously during the progress of stem developmental stages. In contrast, compared with the S1 stage, LsWRKY51 showed decreased expression levels at S2, S3, and S4 stages.

Figure 12 Expression profiles of LsWRKYIII genes at different lettuce stem enlargement periods.

Bars with different lowercase letters were significantly different by Duncan’s multiple range tests at 0.05 levels. S1, diameter length is one cm; S2, diameter length is two cm; S3, diameter length is three cm; S4, diameter length is four cm.

Discussion

Identification of WRKY TFs family in lettuce

The WRKY TFs have been confirmed to participate in various biological processes, including various environmental stresses, plant growth, and development (Dong, Chen & Chen, 2003; Rengasamy et al., 2008; Kouzai et al., 2020; Wei et al., 2021). Because of high-throughput sequencing technology advancement, WRKY TFs family has been identified in numerous higher plants (Ishiguro & Nakamura, 1994; Yang et al., 2020; Qu et al., 2021). Guo et al. (2019) analyzed the WRKY TFs family from Asterales plant orders such as sunflower (Helianthus annuus) and globe artichoke (Cynara cardunculus), and lettuce. There were 112, 60, and 74 WRKY TFs found in sunflower, globe artichoke, and lettuce, respectively. Furthermore, apart from 74 LsWRKY TFs identified by Guo et al. (2019), 2 TFs (LsWRKY02 and LsWRKY38) were also identified as WRKY TFs in lettuce in our study. The difference could be attributed to the different e values used when screening the WRKY domain. Comparative analysis revealed that the number of WRKY TFs members was unrelated to plant genome size. The genome size of Arabidopsis, tomato, and lettuce was 125 Mb, 900 Mb, and 2.5 Gb, respectively, with a similar number of WRKY TFs in Arabidopsis (72), tomato (81), and lettuce (76) (Reyes-Chin-Wo et al., 2017) (Fig. S2). The number of WRKY TFs in both potato and Hevea brasiliensis was 81, but the potato and H. brasiliensis genome sizes were 844 Mb and 2.15 Gb, respectively. These results indicated that the plants’ genome size could not determine the numbers of WRKY TFs.

Phylogenetic analysis of LsWRKY TFs

Many studies showed that genes that belonged to the same subfamily played similar roles (Ding et al., 2015; Ma et al., 2017; Yang et al., 2021). The CaWRKY30, a homolog of AtWRKY30, showed similar roles to AtWRKY30; these 2 WRKY genes positively regulate biotic and abiotic stresses (Scarpeci et al., 2013; El-Esawi et al., 2019; Hussain et al., 2021). To analyze the possible function of LsWRKY TFs, WRKY TFs from Arabidopsis, tomato, and lettuce were chosen to conduct the phylogenetic relationship. As shown in Fig. 2, 76 LsWRKY TFs were classified into seven subfamilies (I, IIa-IIe, and III). AtWRKY23 and AtWRKY12 were identified to regulate embryo development and secondary cell wall formation (Wang et al., 2010; Grunewald et al., 2013). As the homology of AtWRKY23 and AtWRKY12, LsWRKY46 and LsWRKY05 might play similar roles in the progress of embryo development. Liu et al. (2012) found that WRKYII subfamily TFs (AtWRKY18, AtWRKY40, and AtWRKY60) were involved in the ABA signaling pathway. Thus, it is envisaged that the homology of AtWRKY18, AtWRKY40, and AtWRKY60, 3 WRKYII TFs (LsWRKY13, LsWRKY34, and LsWRKY28) in lettuce may also be involved in ABA signaling pathways. However, the function still needs further verification. Compared with WRKYI and WRKYII, WRKYIII subfamily TFs were proved as the most adaptable and advanced in monocot. The proportion of WRKYIII TFs ranged from 10% to 36% (Fig. S3). The 14 WRKYIII members in Arabidopsis participate in different plant defense signaling pathways, indicating that WRKYIII evolution needed increasing biological requirements (Kalde et al., 2003).

Roles of LsWRKY TFs in abiotic stress

Many studies have confirmed the role of WRKY TFs in plant growth and development, pathogen defense, and abiotic stress (Dong, Chen & Chen, 2003; Rengasamy et al., 2008; Kouzai et al., 2020; Wei et al., 2021). A variety of abiotic factors such as drought and salt could induce the expression of WRKY genes. In Arabidopsis, 26 WRKY TFs were identified to respond to abiotic stress (Jiang & Deyholos, 2006). Similarly, most WRKYIII subfamily genes participate in abiotic stress. The AtWRKY30 improved resistance to salt stress and oxidative stress (Scarpeci et al., 2013), while AtWRKY46 overexpression plants showed more sensitivity to drought and salt stress (Ding et al., 2015). Similar results were found in wheat. TaWRKY146, a homolog of AtWRKY46, also negatively regulated drought and salt stresses (Ma et al., 2017). OsWRKY76 improved the resistance of rice to cold stress (Naoki et al., 2013). Our results showed that different expression patterns of LsWRKYIII genes responded to abiotic stresses. Drought could induce the expression of LsWRKY12, LsWRKY39, LsWRKY51, LsWRKY75. The expression levels of most LsWRKYIII genes, except LsWRKY58, decreased under salt treatment for 24 h. LsWRKY58, the homology of AtWRKY30, showed increased expression levels under salt treatment for 24 h, which indicated that LsWRKY58 might play a similar function to AtWRKY30 in response to salt stress (Scarpeci et al., 2013).

Roles of LsWRKY TFs in plant growth and development

Some valuable clues have revealed the roles of WRKY TFs in plant development, including seed development, senescence, seed dormancy, and germination (Sun et al., 2003; Luo et al., 2005; Zhou, Jiang & Yu, 2011). TTG2, one of the WRKY TFs, was identified to play a role in organ development for the first time (Johnson, Kolevski & Smyth, 2002). Gene expression profiles are linked to gene function (Xu et al., 2015). Rice OsWRKY78 was confirmed to promote seed development and stem elongation (Zhang et al., 2011). AcWRKY TFs may also play a role in specific pineapple physiological processes (Xie et al., 2018). Ding et al. (2015) found that AtWRKY46 positively regulated the lateral root development. WRKYIII gene GhWRKY53, an orthologous gene of AtWRKY46, could significantly increase the density of trichomes in A. thaliana and increase the yield of cotton fiber (Yang et al., 2021). These studies cemented the critical role of WRKYIII subfamily TFs in plant’s growth and development. RNA-Seq analysis revealed that 43 of 76 LsWRKY TFs had different expression patterns in different stages of stem enlargement (Fig. 11). The expression levels of the WRKYIII subfamily in response to stem enlargement were identified by qRT-PCR. Most genes except LsWRKY51, LsWRKY69, and LsWRKY72, showed increased expression during the progress of stem enlargement, indicating that differently expressed WRKYIII TFs may be the key regulators of lettuce stem development (Fig. 12).

Regulation mechanism of WRKY TFs involved in different biological processes

The regulatory mechanisms of WRKY TFs in plant biological progress are complex. WRKY TFs can effectively combine to W-box existing in the promoter regions of downstream target genes to regulate the expression of target genes or bind other acting elements to form protein complexes. In brief, WRKY TFs could work in coordination or independently in response to different biological processes.

W-box elements exist in many TFs, including WRKY TFs. Hence WRKY TFs can combine with the W-box element in other WRKY TFs to form self-regulation or cross-regulation networks (Li et al., 2020b; Zentgraf, Laun & Miao, 2010). For example, AtWRKY57 played a positive role in drought stress response by binding W-box elements that existed in the promoter of drought-resistant gene RD29A and NCED3 (Jiang, Liang & Yu, 2012). Similarly, SbWRKY50 from Sorghum bicolor participated in salt response by directly binding the promoters of SOS1 and HKT1 (Song et al., 2020). While, AtWRKY34 played negative roles in the CBF-mediated cold response pathway (Zou, Jiang & Yu, 2010). The WRKY TFs binding site W-box element (C/TTGACT/C) was found in many LsWRKY TFs, including LsWRKY03, LsWRKY06, LsWRKY14, LsWRKY20, and LsWRKY36, which indicated that these genes might participate in different biological processes by self-regulation or cross-regulation with other WRKY TFs. Similarly, WRKY TFs also could improve tolerance to various abiotic stresses by increasing some material accumulation. For instance, the overexpression of Boea hygrometrica BhWRKY1 in Nicotiana tabacum improved the seedling drought resistance by inducing the accumulation of raffinose family oligosaccharides (Wang et al., 2009).

WRKY TFs may have crosstalk with plant hormones to avoid different stress conditions (Negi & Khurana, 2021; Lim et al., 2022). The function of WRKY TFs in abiotic stress is often related to defense-associated phytohormones such as JA, SA, and ABA. As a major phytohormone, ABA has been shown to increase salt and drought tolerance (Yin et al., 2017). The ABA could improve drought tolerance by attenuating the inhibition of OsWRKY5 to its downstream gene, such as OsMYB2 (Lim et al., 2022). Chrysanthemum morifolium CmWRKY1 participated in drought response by an ABA-mediated pathway (Fan et al., 2016). In addition to ABA, WRKY TFs play an essential role in the SA signaling pathway. AtWRKY39 responded to high temperatures by collaboratively participating in SA and JA signaling pathways (Li et al., 2010). According to Kim et al. (2008), AtWRKY38 and AtWRKY62 inhibited the expression of the SA responsive gene AtPR1 and decreased tolerance to pathogens. Our study identified hormone-responsive elements (SA, GA, ABA, and auxin) in LsWRKY TFs. The GA-responsive element existed in the promoter of 9 LsWRKYIII TFs except for LsWRKY39; the ABA-responsive element also existed in the promoter of 9 LsWRKYIII TFs except for LsWRKY51; the promoter region of LsWRKY09, LsWRKY51, and LsWRKY75 had SA-responsive element. As shown in Fig. 9, most LsWRKYIII TFs could participate in plant hormone responses (ABA, SA, and GA) with different expression patterns. All the results indicated that LsWRKY TFs, including WRKYIII TFs, may interplay with plant hormones to enhance the adaptation to stress conditions.

LsWRKY TFs could interact with other proteins such as WRKY TFs, MPK, and SIB to regulate different biological processes (Fig. 7). MAPK cascades, as an important signal transduction pathway, play vital roles in the progression of plant disease resistance (Horak, 2020; Yao et al., 2020). WRKY TFs can be phosphorylated and activate MAPK, triggering downstream signaling pathways (Chi et al., 2013; Yao et al., 2020). After being directly phosphorylated by MPK3 and MPK6, AtWRKY33 played a significant role in the progression of fungus-induced camalexin accumulation (Mao et al., 2011). Yao et al. (2020) confirmed that WRKY TFs induced a critical defense response in tobacco resistance to whitefly after being phosphorylated by MAPK. The VQ proteins (containing VQ motif FxxhVQxhTG) as a class of plant-specific transcriptional regulators could fine-tune plant growth or stress regulatory networks by cooperating with their interacting partners, including WRKY TFs (Lai et al., 2011; Hu et al., 2013). Lai et al. (2011) found that VQ proteins, SIB1 and SIB2 could serve as transcriptional activators of WRKY33 in response to Botrytis cinerea. Hu et al. (2013) identified that VQ9, in collaboration with WRKY8, can regulate the plant-salt stress response. The results showed a complex regulation mechanism of LsWRKY TFs during various growth conditions, which may form complexes with other proteins such as SIB, MPK, and TFs.

Conclusion

In general, 76 WRKY TFs were identified in lettuce in the present study. A comprehensive analysis of LsWRKY TFs was conducted, including structural characteristics, phylogenetic relationships, chromosomal distribution, interaction network, and expression profiles. qRT-PCR analysis indicated that the LsWRKYIII genes could respond to abiotic stress, hormone treatment, and stem enlargement. This study provides a theoretical basis for enriching WRKY TFs to regulate stem enlargement and for further exploration of the function of plant WRKY members.

Supplemental Information

Figure S1 Alignment of the amino acid sequence of LsWRKY TFs

The red box represented the conservative domain WRKYGQK of WRKY TFs; amino acids of different colors indicated different degrees of similarity.

Click here for additional data file.

Figure S2 WRKY family TFs members among different plant species

Different colored boxes represented different WRKY subfamily TFs.

Click here for additional data file.

Figure S3 Phylogenetic analysis of WRKY TFs lettuce and Arabidopsis by MEGA7.0

Different colors represent different subfamily WRKY TFs.

Click here for additional data file.

Supplemental Information 1 Supplemental tables

Click here for additional data file.

Additional Information and Declarations

Competing Interests

Author Contributions

Data Availability

The authors declare there are no competing interests.

Ping Du performed the experiments, prepared figures and/or tables, authored or reviewed drafts of the article, and approved the final draft.

Qinglian Wu performed the experiments, authored or reviewed drafts of the article, and approved the final draft.

Yihua Liu performed the experiments, authored or reviewed drafts of the article, and approved the final draft.

Xue Cao analyzed the data, prepared figures and/or tables, authored or reviewed drafts of the article, and approved the final draft.

Wenyi Yi analyzed the data, authored or reviewed drafts of the article, and approved the final draft.

Tikun Jiao performed the experiments, authored or reviewed drafts of the article, and approved the final draft.

Mengqi Hu analyzed the data, prepared figures and/or tables, authored or reviewed drafts of the article, and approved the final draft.

Ying Huang conceived and designed the experiments, authored or reviewed drafts of the article, and approved the final draft.

The following information was supplied regarding data availability:

The raw measurements are available in the Supplemental Files.

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
