# Peer review of "WRKY transcription factor family in lettuce plant (Lactuca sativa): Genome-wide characterization, chromosome location, phylogeny structures, and expression patterns"

_PeerJ, doi:10.7717/peerj.14136_

## Round 0.1 · original submission · Major Revisions

In the present form, the article can not be accepted. Extensive work is required to improve the articles. Kindly incorporate the recommended changes.

·

Basic reporting

no comments

Experimental design

seems good but the authors may have performed the statistical analysis of the data for which they have given the bar graphs.

Validity of the findings

no comment

Additional comments

The manuscript titled “WRKY transcription factor family in lettuce plant (Lactuca sativa): Genome-wide characterization, chromosome location, phylogeny, structures, and expression patterns” was submitted for publication in the PeerJ. In this manuscript, the authors have identified and analyzed WRKY genes in lettuce using different analyses like phylogenetic analysis, analysis of cis-elements in the promoter chromosomal positions of genes, expression analysis in various plant parts, in response to hormones, abiotic stresses, and developmental stages of the stem. Moreover, some WRKY genes showed differential expression in response to different abiotic stresses like cold, salt, and drought. The manuscript reports some novel work in the domain, however, it cannot be published in its present form in the journal. Following issues must be addressed before the acceptance of the manuscript.
1. The English language is not up to the mark. Drastic improvement in the English language is required. It is better to get the services of some editing agencies to improve language and sentence structure.
2. What do the authors want to demonstrate by showing a single cross-section of the stem? The authors should show the development of the stem by showing cross-sections of different stages of the stem. The stages for they have shown the expression of WRKY genes in Figure 10.
3. For identification of WRKY genes authors used SMART. Was it used twice as written in the material and method section?
4. The other major problem is the resolution of the Figures. For instance, in figure 3, the domain colours, structure of introns and exons and colours of cis-elements are not clear.
5. Moreover, figure captions need huge improvement as the figures must be self-explanatory. The authors should improve the captions by adding some words to make the figures self-explanatory.
6. Similarly, in figure 10, different stem enlargement periods i.e. S1a to S4c are not defined in the figure legend.
7. The interaction networks of LasaWRKYs are not described in the required details in the results section.
8. The identification of WRKY genes and proteins from lettuce does not seem rigorous as the authors do not mention the duplicate proteins which may arise from alternative splicing.
9. Strange that most of the genes were down-regulated in the stem for which the qPCR expression was measured (Figure 9). The authors say that they might be negatively involved in stem enlargement. Have the authors developed some mutants and overexpression lines to prove that?

Reviewer 2 ·

Basic reporting

no comment

Experimental design

no comment

Validity of the findings

no comment

Additional comments

In the present study, authors identified 76 WRKY TFs and constructed the
analysis of structural characteristics, phylogenetic relationships, chromosomal distribution, and their expression profiles in asparagus lettuce. These results provide systematic and comprehensive information on WRKY TFs and lay the foundation for further clarification of the regulatory mechanism of WRKY TFs involved in stress response and the progression of plant growth and development. It is interesting manuscript, but there are some issues that should be addressed.

Major concerns
1. For the phylogenetic tree, I suggest construct a tree using only WRKYs from Arabidopsis and asparagus lettuce(in supplemental material) . The functions of most of the Arabidopsis WRKY are known, which are helpful to predict the function of lettuce WRKYs. The transcription factor from the same subgroup always plays a similar role in a in a specific process, which already have been described in many papers. For example, authors determined the expression of 5 WRKY genes under salt stress. As authors described , AtWRKY30 play a role in salt response, I think it is better to check the expression of wrky genes in the same subgroup of AtWRKY30…



“WRKY transcription factors and plant defense responses: latest discoveries and future prospects”
“MYB transcription factors in Arabidopsis”,
“Transcriptional integration of plant responses to iron availability”
“The transcriptional control of iron homeostasis in plants: a tale of bHLH transcription factors?”
“Regulatory Mechanisms of bHLH Transcription Factors in Plant Adaptive Responses to Various Abiotic Stresses”

2. For the discussion part, especially for the “The functions of LasaWRKY TFs”, Line 384-435.
As describe above, authors should discuss the potential function of the lasaWRKY based on the phylogenetic tree besides the gene expression. For example, AtWRKY23 and AtWRKY12 play a role in embryo development and secondary cell wall formation, respectively . AtWRKY18, AtWRKY40 and AtWRKY60 are involved in the ABA signaling pathway. So, please clarify the closest homolog of these WRKYs in asparagus lettuce. Discuss whether these TFs play a role in the same metabolic process. The discussion part needs a lot of revision.

3. Why authors chose the five WRKYs (LasaWRKY16, LasaWRKY32, LasaWRKY39, LasaWRKY55 and LasaWRKY58) for further expression analysis under different stress? Because they showed high expression levels across the stem developmental stages? I suggest choosing some genes for analysis according to the phylogenetic tree.
4. Line290-293: the five WRKY genes contains hormone-responsive elements in their promoter regions? Again, the gene selection for expression pattern analysis is a problem. Authors showed a lot of data here, but they did not tell a good story. Please manage these data carefully.


Minor concerns:
1. Line 18: “Because of its high nutrient content, asparagus lettuce plays an important role in balancing people's diets.” This sentence should be rewritten

2. Line 56-69, I suggest remove this part. The topic of this manuscript is related to the WRKY. Why authors introduced too much other TFs here. The WRKY TFs are involved in tuber development and stress response. I think the description in lines 70-96 are enough.

3. The quality of Figure 3 is so bad. The Figures 7, 8 have same problem

Reviewer 3 ·

Basic reporting

In the current study, authors have identified WRKY transcription factors by analyzed the genome sequence of L. sativa (available online) and tried to validate the function of some of the WRKY TFs in abiotic stresses, hormone signaling and stem development.
The major drawback of the study is that
1) most of the TFs reported in the current study were already been reported by Guo et al., 2019. Hence it is highly recommended that authors acknowledge the previous studies and use the same nomenclature that was used in earlier publication. Also, authors need to focus only on the new TFs found in their current study and not reported elsewhere. Authors also have to make necessary corrections with correct nomenclature of the TFs reported earlier wherever necessary.
2) Authors mentioned that they performed the sequencing of the transcriptome but have not provided enough details of the experiments. Also, the deposit ID of the sequences were not provided.
3) Although there were other WRKY TFs that were highly upregulated during different stem developmental stages, it was not clear why only 5 TFs that showed lower or differential expression patterns across different stages were selected for functional validation. Please justify.

Experimental design

Authors are requested to provide sufficient statistical analysis of the results obtained in their current study. In some sections, the results described in the manuscript does not match the figures provided. So, authors are advised to review their manuscript thoroughly and make necessary corrections suggested in the annotation version of the manuscript.

Validity of the findings

Discussion section should discuss more about the current study and should be supported by earlier studies and the authors have to provide a conclusion statement describing the final outcome of their study in the form of a paragraph.

Annotated reviews are not available for download in order to protect the identity of reviewers who chose to remain anonymous.

---

## Round 0.2 · Minor Revisions

Although the majority of the points suggested by the reviewers have been incorporated by the authors but some points still need to be addressed. Kindly improve the article as suggested by the reviewers e.g.
1. Authors have provided the RNA-Seq submission ID but it could not be verified in the NCBI. So authors are advised to recheck and provide the appropriate details.
2. Although authors have included the statistical significance in the figures, they have not mentioned the statistical method/s used in the appropriate sections.
3. It is still not clear why only 10 WRKYIII genes were selected for qRT-PCR analysis. Authors are advised to provide a reasonable argument for this.
4. In one of the figures, two new TF's identified in their study was not included.

Reviewer 2 ·

Basic reporting

all my suggestions have been addressed. thanks

Experimental design

ok

Validity of the findings

ok

Reviewer 3 ·

Basic reporting

Authors have improved their manuscript as per the reviewers suggestions and comments. However there are few comments that need to be addressed appropriately. As suggested in the earlier review report,
1. Authors have provided the RNA-Seq submission ID but it could not be verified in the NCBI. So authors are advised to recheck and provide the appropriate details.
2. Although authors have included the statistical significance in the figures, they have not mentioned the statistical method/s used in the appropriate sections.
3. It is still not clear why only 10 WRKYIII genes were selected for qRT-PCR analysis. Authors are advised to provide a reasonable argument for this.
4. In one of the figures, two new TF's identified in their study was not included.
All other additional suggestions and comments are included in the edited version of the revised manuscript. Authors are advised to go through them and provide their response appropriately.

Experimental design

No comments

Validity of the findings

No comments

Annotated reviews are not available for download in order to protect the identity of reviewers who chose to remain anonymous.

---

## Round 0.3 · accepted · Accept

The authors have incorporated all the changes recommended by reviewers. The manuscript is acceptable for publication.

Reviewer 3 ·

Basic reporting

Authors have incorporated all the recommendations and addressed the comments appropriately. The manuscript can be accepted for publication.

Experimental design

No comments

Validity of the findings

No comments